# Determinants of glycemic control among persons living with type 2 diabetes mellitus attending a district hospital in Ghana

**Alexander Adjei** [1,2]*, **Kennedy Tettey Coffie Brightson**[1], **Michael Matey Mensah**[1], **Jemima Osei**[1], **Moses Drah**[1], **Clement Tetteh Narh**[3], **Kwabena Asare**[4], **Francis Anto**[5]

**1** Shai-Osudoku District Hospital, Dodowa, Ghana, **2** Department of Epidemiology, Dodowa Health Research Centre, Dodowa, Ghana, **3** Department of Epidemiology and Biostatistics, Fred N. Binka School of Public Health, University of Health and Allied Sciences, Ho, Ghana, **4** Department of Non-Communicable Disease Epidemiology, London School of Hygiene and Tropical Medicine, London, United Kingdom, **5** Department of Epidemiology and Disease Control, School of Public Health, College of Health Sciences, University of Ghana, Accra, Ghana

* caesar306@yahoo.com

**Data Availability Statement:** All relevant data are within the paper and its Supporting Information files.

## Abstract

### Background

Diabetes mellitus is a growing public health emergency with prevalence in sub-Sahara Africa expected to experience the highest increase by 2045. Glycemic control is central to diabetes management, but it is influenced by various factors. This study determines the level of glycemic control and the associated individual factors among type 2 diabetes mellitus (T2DM) patients.

### Methods

A cross-sectional descriptive study was conducted at the Shai-Osudoku District Hospital from 9th November to 15th December 2022. A structured questionnaire was used to collect data on socio-demographic characteristics, lifestyle modifications, co-morbidities, adherence to medication and diet regimens and duration of diabetes. Anthropometric and glycated hemoglobin (HbA1c) measurements were taken. Chi-squared and multivariate logistic regression analyses were carried out to determine factors associated with glycemic control at 95% confidence levels.

### Results

A total of 227 patients participated in this study. The majority of the participants were females (77.97%) and the mean ($\pm$SD) age was 60.76 $\pm$ 12.12 years. Good glycemic control (HbA1c < 7%) among the participants was 38.77% (n = 88) and the median HbA1c was 7.5% (IQR: 6.5% to 9.4%). Significant factors associated with good glycemic control were eating healthy meals (AOR: 4.78, 95% CI: 1.65, 13.88: p = 0.004), oral hypoglycemic agents alone (AOR: 15.71, 95% CI: 1.90, 129.44: p = 0.010) and those with previously good glycemic control (AOR: 4.27, 95% CI: 2.16, 8.43: <0.001).

**Funding:** The author(s) received no specific funding for this work.

**Competing interests:** The authors have declared that no competing interests exist.

## Conclusion

This study showed low levels of good glycemic control among T2DM patients at the primary care level in Ghana. Healthy eating, oral hypoglycemic agents and those with previously normal HbA1c were associated with glycemic control.

## Introduction

The increasing burden of non-communicable diseases especially Type 2 diabetes mellitus (T2DM) has become a public health emergency [1]. The estimated 10.5% global prevalence of diabetes among 20-79-year-olds in 2021 is expected to increase to 12.2% by 2045. Middle-income countries are expected to experience the highest relative increase compared to high and low-income countries over this period. The African region had the lowest estimated prevalence of 4.5% in 2021 but is expected to witness the highest relative increase in persons living with diabetes by 2045 [2]. The prevalence of T2DM in Ghana was estimated to be 6.46% among adults in 2018 [3], much higher than the African average.

Glycemic control remains the basis of diabetes management but evidence shows poor levels of control worldwide with an even higher burden in low and middle-income countries [4–10]. Studies in Ghana have reported the prevalence of poor glycemic control to be more than 59% using glycated hemoglobin levels (HbA1c) of 7% and above [11,12] or an average 3-month blood glucose level of above 130 mg/dl [13].

The consequences of this high burden of poor glycemic control are macro and microvascular complications such as retinopathy, nephropathy, stroke and ischemic heart diseases [14]. These lead to high morbidity with poor quality of life and mortality as well as the economic burden on these persons and the health system [15,16]. Glycated hemoglobin level which measures the average plasma glucose over the previous three months correlates well with these macro and microvascular complications [17].

Different factors have been demonstrated to influence glycemic control from socio-demographic [18–20] to socio-economic characteristics [11,21–23]. Others factors include co-morbidities and lifestyle modification [11,24–27], dietary and medication adherence [6,18,28,29], type of anti-diabetic medication used by patients [13,30,31] and duration of diabetes [5,**32,33**].

Glycemic control can be a complex and difficult process with different individual factors contributing to it. Knowing these factors in the local context will enhance policy formulation to improve the health outcomes of T2DM patients and prevent complications. This will improve patients' quality of life and decrease the financial burden on the scarce resources of the health sector in low and middle-income countries. This study aimed to assess the glycemic level and categorize the individual factors that influence glycemic control among T2DM patients attending clinic at Shai-Osudoku District Hospital.

## Methods

### Study design

A cross-sectional descriptive study was conducted to determine factors associated with glycemic control among Type 2 diabetes mellitus patients attending the out-patient unit of the Shai-Osudoku District Hospital from 9th November 2022 to 15th December 2022. Patients were sequentially enrolled on the study and face-to-face interviews were conducted. Data on socio-demographic characteristics, lifestyle and history of diabetes were collected from the

patients using a structured questionnaire. Anthropometric and glycated hemoglobin measurements were also done. The patients' clinic records were reviewed and data on previous HbA1c levels and medications being taken extracted.

## Study site

The study was carried out at the Shai-Osudoku District Hospital (SODH), located in Dodowa, the capital of Shai-Osudoku District of the Greater Accra Region, Ghana. It is a 140-bed hospital with in- and out-patient facilities and laboratory services. The hospital has an average out-patient attendance of 60,000 annually and runs a chronic disease care clinic for diabetic and hypertensive patients. It serves the district's population of about 105,600 (according to Ghana's 2020 population and housing census). The hospital also serves as the main referral centre for the lower-level health facilities in the surrounding peri-urban and rural communities. The district has four government health centres, 10 Community-based Health Planning and Services (CHPS) compounds, one private hospital, one mission hospital and four private clinics. The inhabitants are mainly subsistence farmers, fishermen, artisanal workers, craftsmen, and petty traders. There are a few civil servants, mainly migrant employees of government ministries, departments, and agencies.

## Sample size estimation and sampling

Yamane's formula $[(n = N/1 + Ne^2)$ where n = the required sample size, N = the total population and e = the margin of error of 0.05] was used to estimate the required minimum sample size based on the population of 240 visiting the diabetic clinic regularly. Using a 5% margin of error and 5% non-response rate, a sample size of 158 patients was estimated. To increase the statistical power and allow for further sub-group analysis, however, all Type 2 diabetes mellitus patients who consented to be part of the study were sequentially enrolled.

## Inclusion/Exclusion criteria

All Type 2 diabetes mellitus patients who visit the study facility regularly and consented to be part of the study were eligible for participation except those who were seriously ill.

## Data collection tool

A questionnaire was developed specifically for this work, incorporating the Perceived Dietary Adherence Questionnaire (PDAQ) for persons living with Type 2 diabetes mellitus and the Morisky Green Levine medication adherence scale. The questionnaire had four main sections. The first section captured data on the background characteristics of study participants including age, sex, marital status, employment status, level of education, place of residence, household size, sex of head of household, and lifestyle characteristics. The second section captured data on medical history including duration of diabetes, existence of comorbidities and type of antidiabetic medication being taken. The third section collected data on the patient's perceived dietary adherence. This section consisted of nine questions structured to capture a nutritional therapy guideline that was adapted to the food items in our local settings consisting primarily of carbohydrates, fats and oils, fruits and vegetables. The fourth section captured data on medication adherence based on the Morisky Green Levine medication scale. Information collected included whether the patient sometimes forgot to take his or her medications, whether in the past two weeks, there was any day the patient did not take his or her medications and whether the patient took his or her medications the day before the study.

## Data collection procedures

**Interviews and data extraction.** Face-to-face interviews lasting about 20 minutes were conducted with the study participants after their scheduled clinic review using the structured questionnaire. The interviews were done by trained research officers following standard operating procedures. Data were collected on demographic and lifestyle characteristics, comorbidities and duration of diagnosis of diabetes mellitus. Dietary and drug adherence information was also collected from the patients and updated using clinic records as source documents where necessary.

**Weight measurement.** Patients were weighed using the Seca dial column weighing scale (JACKERMAC GERMANY) with an attached stadiometer placed on an even surface and adjusted to zero. They were weighed standing straight on the scale completely with weight evenly distributed between their feet placed in the center of the scale, not touching anything, without shoes and the pockets emptied. The recorded measurement was checked for accuracy and weight measured to the nearest 100g.

**Height measurement.** Patients stood straight with feet together on the stadiometer without shoes. The patient's knees, back and neck were straightened, back close to the scale and the head lever placed on top of the head. The height of the patient was measured from the level recorded from the scale by the study nurse. Height was measured to the nearest 0.1cm and recorded values were checked for accuracy.

**Assessing HbA1c.** A point-of-care calibrated automated analyzer (A1 Care$^{TM}$) was used to estimate the HbA1c levels. The automated HbA1c analyzer uses cartridge which has a sample collection device and a lysing buffer cell. The sample collector which has a handle was removed from the cartilage and used to collect about 2.5μls of anti-coagulated venous blood from an EDTA tube. The sample collector was gently placed and positioned inside the cartilage which was then completely inserted into the reaction chamber of the analyzer. The sample collector handle was gently pressed down to rupture the lysing buffer and the reaction chamber closed. The sample was analyzed and the HbA1c results displayed within 5 minutes.

## Data management and analysis

All completed questionnaires were checked for missing information and errors daily. The data were entered and cleaned using Epi Info Version 7. Statistical analyses were done using STATA version 17. Descriptive statistics were used to present data as frequencies with their corresponding percentages. The proportion of patients with good glycemic control (HbA1c <7%) was determined and its distribution across the independent variables was presented in cross-tabulations.

The weight and height were used to calculate the body mass index (BMI) of patients. Weight in kilograms was divided by the square of the height in meters. BMI was categorized as underweight (<18.5kg/m$^2$), normal weight (18.5–24.9kg/m$^2$), overweight (25.0–29.9kg/m$^2$), and obese (>30.0kg/m$^2$). Medication adherence was assessed based on four main questions with Yes or No responses. The questions were negatively coded. The lower the total score, the more adherent the patient was to the medication regimen. This was categorized as low (score of 3–4), moderate (score of 1–2) and high (score of 0) adherence.

Spearman rank correlation was used to determine the relationship between glycemic control and continuous variables. The Chi-square test was used to determine the association between glycemic control and categorical variables. One-way analysis of variance was done for categorical variables with more than two groups. Fisher's exact test and Wilcoxon rank-sum test were used as non-parametric measures. A p-value of less than 0.05 was used to select individual-level factors from the Chi-square test for the multivariate analysis. Multivariate logistic

regression was conducted to determine the strength of association between the individual-level factors and glycemic control at 95% confidence level.

## Ethical consideration

The study protocol, data collection tool and consent form were reviewed and approved by the Ethics Review Committee (GHS-ERC) of the Ghana Health Service with Protocol ID GHS-ERC 044/09/22. Written informed consent was obtained from all study participants before enrollment into the study, and the confidentiality of patient identity and information was respected. The study was conducted under Good Clinical Practices and Helsinki Declaration for biomedical research on human subjects.

## Results

### Demographic and clinical characteristics of patients

A total of 230 Type 2 diabetes mellitus patients visited the clinic during the study period and consented to participate in the study. Out of the 230 patients screened, 227 (98.70%) met the inclusion criteria and were enrolled and included in the analyses. The mean age of the enrolled patients was 60.76 ($\pm$ 12.12) years with the majority of them being female (77.97%). Most of the patients (62.11%) were married and 77.54% had completed secondary or higher education. One hundred and fourteen (50.22%) of them were employed and 69.16% lived in urban parts of the district. The majority of the patients (71.11%) were either overweight or obese. Almost 60% of the patients have been living with diabetes mellitus for five or more years and over 74% do not exercise at all or exercise only once in a while. Over 85% were on oral hypoglycemic agents alone and more than half of the patients were adhering to medication based on the Morisky Green Levine's medication adherence score (Table 1).

### Level of glycemic control among T2DM patients

Most of the patients (61.23%, 139/227) had poor glycemic control (HbA1c > 7%). The measured HbA1c of patients were positively skewed with a median HbA1c of 7.5% (IQR: 6.5% - 9.4%) as shown in Fig 1. There was no statistically significant difference in the median HbA1c among males and females using the Wilcoxon rank-sum test (p = 0.2402). There was an inverse significant relationship between HbA1c and the age of patients from the Spearman rank correlation (r = −0.2205, p <0.001). There was a statistically significant difference comparing HbA1c by the type of anti-diabetic agents (p<0.001) based on one-way analysis of variance (ANOVA).

### Association between glycemic control and socio-demographic characteristics

A higher proportion of patients (70.75%) aged below 60 years had poor glycemic control compared to those aged 60 years and above (52.89%). A lower proportion (49.54%) of patients who usually eat healthy meals had poor glycemic control compared to 67.09% and 82.05% of those who only sometimes eat healthy meals and those who did not often eat healthy meals respectively. Using the perceived dietary adherence scale, 62.22% of those with low level of adherence had poor glycemic control compared to 57.14% of those with high level of adherence.

There was a statistically significant association between glycemic control and the age of patients (p = 0.006), marital status (p = 0.032), the type of anti-glycemic agent (p <0.001), eating healthy meals (p <0.001), and having a previously good glycemic level (p <0.001) based on Chi-square test (Table 2).

**Table 1.  Demographic and clinical characteristics of study patients.**

| Characteristics | Frequency | Percentage (%) |
|---|---|---|
| **Age Group (years)** | | |
| <60 | 101 | 44.49 |
| ≥60 | 126 | 55.51 |
| **Sex** | | |
| Female | 177 | 77.97 |
| Male | 50 | 20.03 |
| **Marital status** | | |
| Single/widowed/divorced | 86 | 37.88 |
| Married/living together | 141 | 62.11 |
| **Level of education** | | |
| No formal/Primary education | 51 | 22.46 |
| Secondary/Tertiary education | 176 | 77.54 |
| **Employment status** | | |
| Employed | 114 | 50.22 |
| Not employed | 70 | 30.84 |
| Retired | 43 | 18.94 |
| **Residence** | | |
| Peri-urban | 39 | 17.18 |
| Rural | 31 | 13.66 |
| Urban | 157 | 69.16 |
| **BMI of patients (N = 225)** | | |
| Underweight | 4 | 1.78 |
| Normal | 61 | 27.11 |
| Overweight | 68 | 30.22 |
| Obese | 92 | 40.89 |
| **Duration of Diabetes (years)** | | |
| <5 | 91 | 40.09 |
| 5–10 | 77 | 33.92 |
| >10 | 59 | 25.99 |
| **Level of exercise** | | |
| No exercise | 68 | 29.96 |
| Not regular/once in a while | 102 | 44.93 |
| Once to five times a week | 57 | 25.11 |
| **Type of anti-diabetic agents** | | |
| Oral hypoglycemic agents | 195 | 85.90 |
| Insulin | 12 | 5.29 |
| Both | 20 | 8.81 |
| **Adherence to medication (MGL)** | | |
| Low adherence | 22 | 10.13 |
| Medium adherence | 76 | 33.48 |
| High adherence | 128 | 56.39 |

BMI–Body mass index, MGL - Morisky Green Levine medication adherence scale.

## Multivariable analysis of significant independent factors

The multivariable logistic regression analysis revealed that patients who usually eat healthy meals (aOR: 4.78, 95% CI: 1.65, 13.88), are on oral hypoglycemic agents alone (aOR: 15.71,

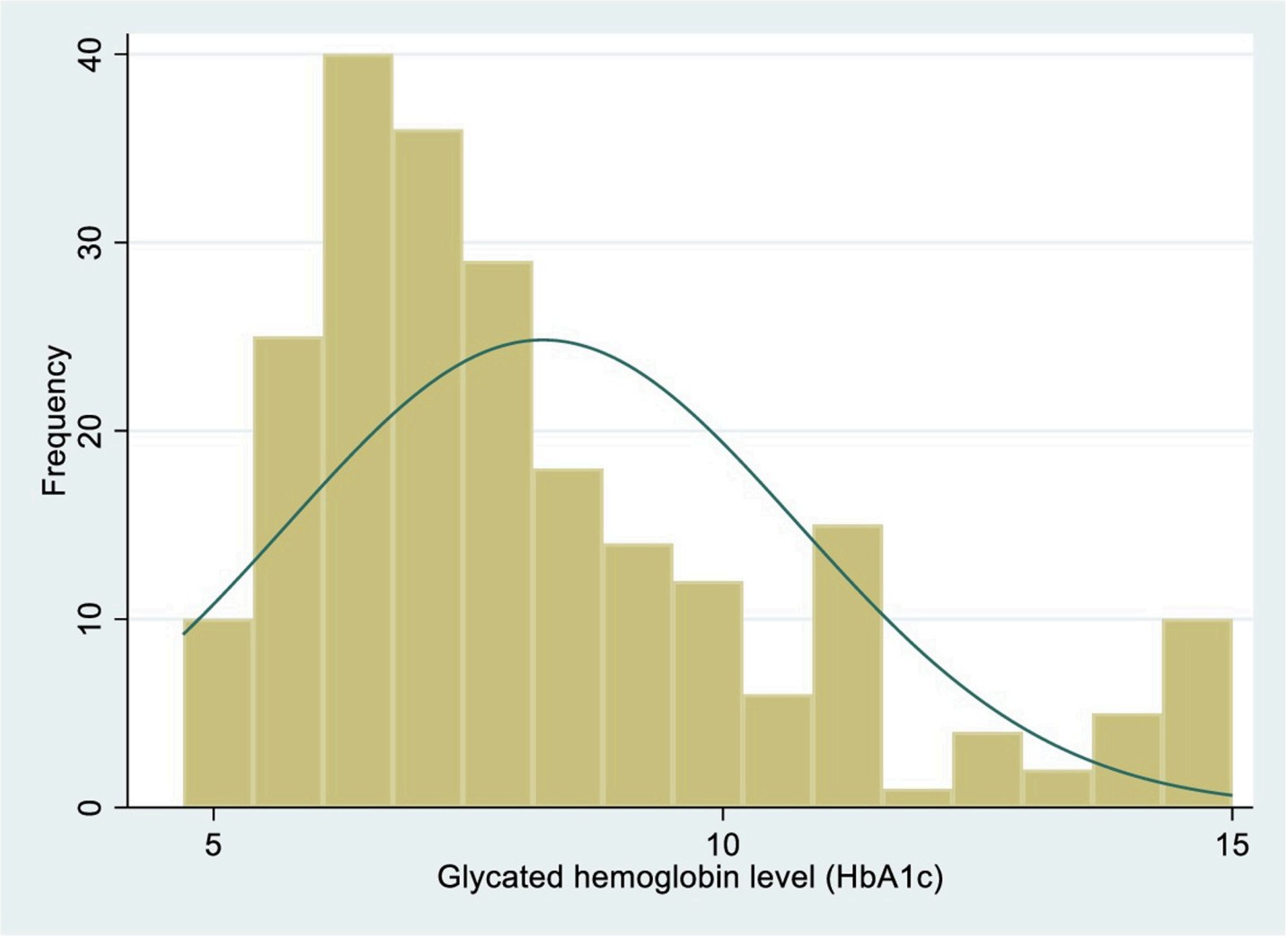

**Fig 1. Graph showing the distribution of glycated hemoglobin of patients.**

95% CI: 1.90, 129.44) and those with previously good glycemic control (aOR: 4.27, 95% CI: 2.16, 8.43) had good glycemic control in this study (Table 3).

## Discussion

This study assessed patient factors associated with glycemic control among type 2 diabetes mellitus (T2DM) patients receiving care at a district hospital in Ghana, West Africa. The study found the level of glycemic control among the patients to be poor (61%), significantly influenced by poor eating habits. The use of insulin alone or in combination with an oral hypoglycemic agent did not appear to improve glycemic control in patients with a history of poor control.

Varying levels of glycemic control have been reported from different studies and among different populations, with the prevalence of poor control ranging from 45–93% among T2DM patients [9]. The high prevalence of poor glycemic control (61%) found in our current study is within the range reported by Bin Rakhis and colleagues in their systematic review and also within the range of 39–69% reported by Swaray and colleagues in Ghana for the period

**Table 2. Association between demographic characteristics and glycemic control.**

| Characteristics | Good Glycemic control (%) | Poor Glycemic control (%) | $\chi^2$ | P - value |
|---|---|---|---|---|
| **Age group (years)** | | | 7.59 | **0.006** |
| <60 | 31 (29.25) | 75 (70.75) | | |
| ≥60 | 57 (47.11) | 64 (52.89) | | |
| **Sex** | | | 0.04 | 0.839 |
| Female | 68 (38.42) | 109 (61.58) | | |
| Male | 20 (40.00) | 30 (60.00) | | |
| **Marital status** | | | 4.63 | **0.032** |
| Single/widow/divorced | 41 (47.67) | 45 (52.33) | | |
| Married/living together | 47 (33.33) | 94 (66.67) | | |
| **BMI (N = 225)** | | | 7.17 | 0.062[a] |
| Underweight | 1 (25.00) | 3 (75.00) | | |
| Normal | 22 (36.07) | 39 (63.93) | | |
| Overweight | 35 (51.47) | 33 (48.53) | | |
| Obese | 29 (31.52) | 63 (68.48) | | |
| **Duration of diabetes** | | | 3.61 | 0.165 |
| Less than 5years | 42 (46.15) | 49 (53.85) | | |
| 5-10years | 27 (35.06) | 50 (64.94) | | |
| More than 10years | 19 (32.20) | 40 (67.80) | | |
| **Type of anti-diabetic agents** | | | 19.93 | **<0.001**[a] |
| Oral hypoglycemic agents | 87 (44.62) | 108 (55.38) | | |
| Insulin/Both | 1 (3.13) | 31 (96.88) | | |
| **Drug adherence (MGL)** | | | 2.31 | 0.312 |
| Low adherence | 6 (26.09) | 17 (73.91) | | |
| Medium adherence | 28 (36.84) | 48 (63.16) | | |
| High adherence | 54 (42.19) | 74 (57.81) | | |
| **Eating healthy meals** | | | 14.54 | **<0.001** |
| Usually | 55 (50.46) | 54 (49.54) | | |
| Sometimes | 26 (32.91) | 53 (67.09) | | |
| Not often | 7 (17.95) | 32 (82.05) | | |
| **Perceived Dietary Adherence Questionnaire scale** | | | 1.56 | 0.212 |
| Low adherence | 40 (34.78) | 75 (62.22) | | |
| High adherence | 48 (42.86) | 64 (57.14) | | |
| **Previous HbA1c** | | | 32.67 | **<0.001** |
| Good control | 47 (67.14) | 23 (32.86) | | |
| Poor control | 38 (26.39) | 106 (73.61) | | |

[a]Fisher's exact test p-value, BMI–Body mass index, MGL - Morisky Green Levine medication adherence scale.

2015–2021 [12]. Similar levels of poor glycemic control, 61% [6,34] and 65% [35], have been reported in Ethiopia. Studies in Nigeria 40% [18], and some industrialized countries, China 50% [27], and the United States of America 28% [36], however reported much lower levels of poor glycemic control.

Poor glycemic control has been associated with several patient and medication factors including having high body mass index and being on insulin therapy. Similarly, some behavioural factors such as high levels of physical activity and eating healthy meals (Djonor, et al, 2021), have been reported to influence good glycemic control. As indicated, the main patient factor found to be associated with good glycemic control in our current study is eating healthy

**Table 3. Multivariable logistic regression for the determinant of glycemic control.**

| Characteristics | Unadjusted | | Adjusted | |
| --- | --- | --- | --- | --- |
| | OR (95% CI) | P-value | OR (95% CI) | P-value |
| **Age group in years** | | | | |
| 60 and below | Ref | | Ref | |
| Above 60 | 2.16 (1.24, 3.74) | **0.006** | 1.37 (0.71, 2.62) | 0.346 |
| **Marital status** | | | | |
| Single/widow/divorced | Ref | | Ref | |
| Married/living together | 0.55 (0.32, 0.95) | **0.032** | 0.55 (0.29, 1.04) | 0.067 |
| **Eating healthy meal** | | | | |
| Not healthy meals | Ref | | Ref | |
| Sometimes | 2.24 (0.87, 5.76) | 0.093 | 1.84 (0.62, 5.44) | 0.271 |
| Healthy meals | 4.66 (1.89, 11.45) | **0.001** | 4.78 (1.65, 13.88) | **0.004** |
| **Type of anti-diabetic agents** | | | | |
| Both/Insulin | Ref | | Ref | |
| Oral hypoglycemic agents | 24.97(3.34, 186.61) | **0.002** | 15.71 (1.90, 129.44) | **0.010** |
| **Previous HbA1c** | | | | |
| Poor control | Ref | | Ref | |
| Good control | 5.70 (3.06, 10.61) | **<0.001** | 4.27 (2.16, 8.43) | **<0.001** |

meals. According to Sainsbury, et al. (2018), a carbohydrate-restricted diet can lead to a reduction in HbA1c and therefore good glycemic control.

Managing T2DM involves a multi-disciplinary team and the role of dieticians is as important as that of clinicians in improving glycemic control [37]. That is, dietary control is an important component of therapeutic lifestyle modification in the management of T2DM [7,25]. Dietary guidelines have therefore been established in terms of carbohydrates, especially fibre-rich, proteins and fats for persons living with T2DM [38,39]. These guidelines need to be individualized in a patient-centered manner taking into consideration the socio-economic implications of such nutritional therapy.

The use of oral hypoglycemic agents alone showed some association with glycemic control in our current study. Previous studies have reported conflicting results with the choice of medication and glycemic control among T2DM. The findings in this study is consistent with an earlier study in Ghana where T2DM patients on insulin therapy were significantly associated with poor glycemic control [11]. Other studies in Greece found the use of injectables (insulin) to be associated with good glycemic control compared to oral hypoglycemic agents alone or in combination with injectables [31]. Ghabban and his colleagues in Saudi Arabia also found the use of combined therapy to be associated with poor glycemic control compared to insulin therapy alone [30].

The use of insulin in T2DM mostly happens late in the management because of the pathophysiology, the relative insulin deficiency against a background of peripheral resistance [40]. The disease progression leads to the gradual loss of the β-cell secretory functions [41], leading to T2DM being reliant on insulin injection in the latter stages to achieve glycemic control. The use of insulin injection comes with low compliance with its consequence on glycemic control [42] and sometimes difficulty in storage especially when dealing with patients with low socio-economic status. Improving compliance through patient education on the proper use of injectables and monitoring HbA1c targets on follow-up visits will help address patient problems, phobias and myths associated with insulin injection.

Another significant factor identified in this study was patients past good glycemic history. This underlines the comprehensive, continuous and coordinated management of T2DM with targeted glycemic control while addressing challenges associated with complications, drug adverse effects and comorbidities.

## Limitations of the study

Glycated hemoglobin levels could be affected by anaemia and hemoglobin variants like sickle cell traits and Glucose-6-phosphate dehydrogenase deficiency. These were not assessed and therefore not controlled for in the analysis. Information on dietary and medication adherence was assessed using adopted tools which could be affected by recall bias. The findings of this study need to be considered in the context of this potential bias due to self-reported data. This is a cross-sectional study and thus causal relationship could not be established. Confounding factors like anemia and hemoglobin disorders may not be evenly distributed, leading to bias and misinterpretation. Multivariate logistic regression analysis helps to control for confounding variables. These limitations notwithstanding the findings of this study can serve as a guide in the management of T2DM patients.

## Conclusion

This study found a high level of poor glycemic control among T2DM patients receiving care at a district hospital in Ghana which was significantly associated with eating unhealthy foods. Strong collaboration among attending clinicians and dieticians could help improve care for such patients.

## Supporting information

**S1 Dataset.**
(XLSX)

## Acknowledgments

The authors wish to acknowledge the management, and staff of Shai-Osudoku District Hospital, especially Dr. Akude, Dr. Majorie Nii Koi, the nurses, and patients at the Diabetic Clinic.

## Author Contributions

**Conceptualization:** Alexander Adjei, Francis Anto.

**Data curation:** Alexander Adjei, Michael Matey Mensah, Jemima Osei, Moses Drah, Clement Tetteh Narh, Kwabena Asare, Francis Anto.

**Formal analysis:** Alexander Adjei, Michael Matey Mensah, Moses Drah, Clement Tetteh Narh, Kwabena Asare.

**Investigation:** Alexander Adjei, Kennedy Tettey Coffie Brightson, Michael Matey Mensah, Jemima Osei, Moses Drah, Francis Anto.

**Methodology:** Alexander Adjei, Clement Tetteh Narh, Kwabena Asare, Francis Anto.

**Project administration:** Alexander Adjei.

**Resources:** Alexander Adjei, Kennedy Tettey Coffie Brightson.

**Software:** Alexander Adjei, Michael Matey Mensah, Moses Drah.

**Supervision:** Alexander Adjei, Francis Anto.

**Validation:** Alexander Adjei, Kennedy Tettey Coffie Brightson, Michael Matey Mensah, Jemima Osei, Moses Drah, Clement Tetteh Narh, Kwabena Asare, Francis Anto.

**Visualization:** Alexander Adjei, Kennedy Tettey Coffie Brightson, Michael Matey Mensah, Jemima Osei, Moses Drah, Clement Tetteh Narh, Kwabena Asare, Francis Anto.

**Writing – original draft:** Alexander Adjei, Kennedy Tettey Coffie Brightson, Kwabena Asare, Francis Anto.

**Writing – review & editing:** Alexander Adjei, Kennedy Tettey Coffie Brightson, Michael Matey Mensah, Jemima Osei, Moses Drah, Clement Tetteh Narh, Kwabena Asare, Francis Anto.

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
