## [Decision Letter · Decision Letter 0]

23 Sep 2024

PONE-D-24-28656Determinants of glycemic control among persons living with type 2 diabetes mellitus attending a district hospital in GhanaPLOS ONE

Dear Dr. ADJEI,

Thank you for submitting your manuscript to PLOS ONE. After careful consideration, we feel that it has merit but does not fully meet PLOS ONE’s publication criteria as it currently stands. Therefore, we invite you to submit a revised version of the manuscript that addresses the points raised during the review process.

We look forward to receiving your revised manuscript.

Kind regards,

Seth Agyei Domfeh, PhD

Academic Editor

PLOS ONE

**Journal Requirements:**

Reviewers' comments:

Reviewer's Responses to Questions

**Comments to the Author**

1. Is the manuscript technically sound, and do the data support the conclusions?

Reviewer #1: Yes

Reviewer #2: Yes

2. Has the statistical analysis been performed appropriately and rigorously? 

Reviewer #1: Yes

Reviewer #2: Yes

3. Have the authors made all data underlying the findings in their manuscript fully available?

Reviewer #1: Yes

Reviewer #2: Yes

4. Is the manuscript presented in an intelligible fashion and written in standard English?

Reviewer #1: Yes

Reviewer #2: Yes

5. Review Comments to the Author

**Reviewer #1: **The the manuscript the authors determines the level of glycemic control and the associated individual factors among type 2 diabetes mellitus (T2DM) patients attending a district hospital.

The authors clearly describes their methods and used appropiate statistical analysis for their data. The manuscript is well written and coherrent. The conclusion reached is supported by the analysis performed.

**Reviewer #2:** REVIEW COMMENTS FOR MANUSCRIPT PONE-D-24-28656

General Comments

Title and Abstract: The title accurately reflects the content of the study. The abstract is well-written, providing a concise summary of the study's objectives, methods, results, and conclusion. However, consider adding more details about the statistical methods in the abstract for clarity.

Introduction: The introduction provides a good background on the global and local burden of diabetes mellitus. The rationale for the study is clearly articulated, and the objectives are well-defined. However, the introduction could benefit from more recent references to strengthen the context of the study.

Methods: The methods section is detailed and appropriate for the research question. The study design, sampling method, and data collection procedures are well-explained. The statistical analysis methods are appropriate, but the handling of potential confounders should be described in more detail.

Results: The results are clearly presented with appropriate use of tables and figures. However, some results are repeated in the text that are already shown in tables, which may be unnecessary. Consider summarizing key points in the text and referring to the tables for detailed results.

Discussion: The discussion appropriately interprets the results in the context of existing literature. The implications of the findings are well-discussed, but the discussion of the study's limitations could be expanded, especially regarding the potential biases in self-reported data and the cross-sectional nature of the study.

Conclusion: The conclusion is concise and highlights the key findings of the study. It appropriately emphasizes the need for improved dietary management among T2DM patients.

References: The references are generally well-chosen and relevant. However, some references are relatively old, and the inclusion of more recent studies could enhance the discussion.

Specific Comments

Clarity and Readability: Some sentences are complex and could be simplified for better readability. For example, the sentence "The cornerstone of diabetes management is glycemic control, a complex process with different contributing factors" could be simplified to "Glycemic control is central to diabetes management, but it is influenced by various factors."

Consistency in Terminology: Ensure that terms such as "glycemic control" and "T2DM" are used consistently throughout the manuscript. For instance, "glycemic control" is sometimes referred to as "blood glucose control" or "HbA1c levels," which could confuse readers.

Statistical Analysis: The choice of statistical tests is appropriate, but the manuscript could benefit from a more detailed explanation of the multivariate logistic regression model, particularly how variables were selected for inclusion in the model.

Figures and Tables: Ensure that all figures and tables are clearly labeled and referenced in the text. Some tables may benefit from additional footnotes explaining abbreviations or providing context.

Limitations: The limitations section is brief and could be expanded to discuss the potential impact of selection bias, the accuracy of self-reported data, and other factors that may have influenced the study's findings.

Recommendations for Improvement

Expand the literature review to include more recent studies that provide context for the findings.

Simplify complex sentences to improve readability and ensure that the manuscript is accessible to a broad audience.

Provide a more detailed explanation of the statistical methods, particularly the logistic regression analysis, to clarify how the results were derived.

Enhance the discussion of limitations to provide a more balanced interpretation of the study's findings.

Ensure all tables and figures are properly labeled and referenced in the text, with clear explanations provided where necessary.

6. PLOS authors have the option to publish the peer review history of their article (what does this mean?). If published, this will include your full peer review and any attached files.

Reviewer #1: No

Reviewer #2: **Yes: **Kofi Boamah Mensah

---

## [Author Response · Author response to Decision Letter 0]

17 Oct 2024

Response to Reviewer #2 comments

General Comments

1. Title and Abstract: The title accurately reflects the content of the study. The abstract is well-written, providing a concise summary of the study's objectives, methods, results, and conclusion. However, consider adding more details about the statistical methods in the abstract for clarity.

Response: Thanks for this comment. We believe that statistical methods in the abstract are appropriate. A detailed description has been provided in the data analysis part of the methods section of the paper.

2. Introduction: The introduction provides a good background on the global and local burden of diabetes mellitus. The rationale for the study is clearly articulated, and the objectives are well-defined. However, the introduction could benefit from more recent references to strengthen the context of the study. 

Response: Five more current papers have been added to this section and these are reference numbers 10, 13, 29, 33 and 34 on pages 4, 5 and 16.

3. Methods: The methods section is detailed and appropriate for the research question. The study design, sampling method, and data collection procedures are well-explained. The statistical analysis methods are appropriate, but the handling of potential confounders should be described in more detail.

Response: Multivariate logistic regression accounts for potential confounders, which the study design does not address. This has been acknowledged under the limitation section of the paper on page 18.

4. Results: The results are clearly presented with appropriate use of tables and figures. However, some results are repeated in the text that are already shown in tables, which may be unnecessary. Consider summarizing key points in the text and referring to the tables for detailed results.

Response: The content on Table 1 provides a summary of the pertinent variables regarding the background of this study. We believe this to be appropriate.

The wording on Table 2 summarizes only three out of ten variables, while that on Table 3 outlines the significant factors derived from the multivariate logistic regression.

5. Discussion: The discussion appropriately interprets the results in the context of existing literature. The implications of the findings are well-discussed, but the discussion of the study's limitations could be expanded, especially regarding the potential biases in self-reported data and the cross-sectional nature of the study.

Response: The potential biases and limitations of the study design have been emphasized in the limitation section on page 18.

6. Conclusion: The conclusion is concise and highlights the key findings of the study. It appropriately emphasizes the need for improved dietary management among T2DM patients.

Response: Thanks for the comment.

7. References: The references are generally well-chosen and relevant. However, some references are relatively old, and the inclusion of more recent studies could enhance the discussion.

Response: Five new references have been added as stated above.

Specific Comments

1. Clarity and Readability: Some sentences are complex and could be simplified for better readability. For example, the sentence "The cornerstone of diabetes management is glycemic control, a complex process with different contributing factors" could be simplified to "Glycemic control is central to diabetes management, but it is influenced by various factors."

Response: Thanks for the comment and the statement has been revised as suggested under the background section of the abstract on page 2. 

2. Consistency in Terminology: Ensure that terms such as "glycemic control" and "T2DM" are used consistently throughout the manuscript. For instance, "glycemic control" is sometimes referred to as "blood glucose control" or "HbA1c levels," which could confuse readers.

Response: Thanks for your comment. We carefully considered the terminologies used to ensure they accurately convey the intended message.

3. Statistical Analysis: The choice of statistical tests is appropriate, but the manuscript could benefit from a more detailed explanation of the multivariate logistic regression model, particularly how variables were selected for inclusion in the model.

Response: This has been done by including the statement "A p-value of 0.05 from the Chi-square test was used for the multivariate logistic regression" under data management and analysis on page 10.

4. Figures and Tables: Ensure that all figures and tables are clearly labeled and referenced in the text. Some tables may benefit from additional footnotes explaining abbreviations or providing context.

Response: Footnotes have been added to tables 1 and 2 on BMI and MGL as follows; “BMI - Body mass index, MGL - Morisky Green Levine medication adherence scale” on pages 12 and 14.

5. Limitations: The limitations section is brief and could be expanded to discuss the potential impact of selection bias, the accuracy of self-reported data, and other factors that may have influenced the study's findings.

Response: The limitation section has been emphasized to reflect these raised comments on recall bias as follows; “The findings of this study need to be considered in the context of this potential bias due to self-reported data”.

On possible confounders, as follows; “Confounding factors like anemia and hemoglobin disorders may not be evenly distributed, leading to bias and misinterpretation. Multivariate logistic regression analysis helps to control for these confounding variables” on page 18.

Thanks for the recommendations.

---

## [Editor Report · Decision Letter 1]

6 Nov 2024

Determinants of glycemic control among persons living with type 2 diabetes mellitus attending a district hospital in Ghana

PONE-D-24-28656R1

Dear Dr. ADJEI,

We’re pleased to inform you that your manuscript has been judged scientifically suitable for publication and will be formally accepted for publication once it meets all outstanding technical requirements.

Kind regards,

Seth Agyei Domfeh, PhD

Academic Editor

PLOS ONE

---

## [Editor Report · Acceptance letter]

15 Nov 2024

PONE-D-24-28656R1 

PLOS ONE

Dear Dr. Adjei, 

I'm pleased to inform you that your manuscript has been deemed suitable for publication in PLOS ONE. Congratulations! Your manuscript is now being handed over to our production team.

Kind regards, 

on behalf of

Dr. Seth Agyei Domfeh 

Academic Editor

PLOS ONE